cognition/behaviour/evolution

A-not-B error, animal welfare, feeding ecology, perseveration, spatial cognition

**Authors for correspondence:**
C. M. C. Raoult
e-mail: camille.raoult@landw.uni-halle.de
C. Nawroth
e-mail: nawroth.christian@gmail.com

# Goats show higher behavioural flexibility than sheep in a spatial detour task

C. M. C. Raoult[1,2], B. Osthaus[3], A. C. G. Hildebrand[4], A. G. McElligott[5] and C. Nawroth[4,6]

[1]Animal Husbandry and Animal Ecology, Institute of Agricultural and Nutritional Sciences, Martin-Luther University of Halle-Wittenberg, 06120 Halle, Germany
[2]Centre for Proper Housing of Ruminants and Pigs, FSVO, Agroscope Tänikon, 8356 Ettenhausen, Switzerland
[3]School of Psychology and Life Sciences, Canterbury Christ Church University, Canterbury CT1 1QU, UK
[4]Biological and Experimental Psychology, School of Biological and Chemical Sciences, Queen Mary University of London, London E1 4NS, UK
[5]Jockey Club College of Veterinary Medicine and Life Sciences, City University of Hong Kong, Kowloon, Hong Kong SAR, People's Republic of China
[6]Institute of Behavioural Physiology, Leibniz Institute for Farm Animal Biology, 18196 Dummerstorf, Germany

CMCR, 0000-0001-5587-108X; BO, 0000-0001-8835-3190; AGM, 0000-0001-5770-4568; CN, 0000-0003-4582-4057

The ability to adapt to changing environments is crucial for survival and has evolved based on socio-ecological factors. Goats and sheep are closely related, with similar social structures, body sizes and domestication levels, but different feeding ecologies, i.e. goats are browsers and sheep are grazers. We investigated whether goats' reliance on more patchily distributed food sources predicted an increased behavioural flexibility compared to sheep. We tested 21 goats and 28 sheep in a spatial A-not-B detour task. Subjects had to navigate around a straight barrier through a gap at its edge. After one, two, three or four of these initial A trials, the gap was moved to the opposite end and subjects performed four B trials. Behaviourally more flexible individuals should move through the new gap faster, while those less behaviourally flexible should show greater perseveration. While both species showed an accuracy reduction following the change of the gap position, goats recovered from this perseveration error from the second B trial onwards, whereas sheep did so only in the fourth B trial, indicating differences in behavioural flexibility between the species. This higher degree of flexibility in goats compared to sheep could be linked to differences in their foraging strategies.

# 1. Background

Animals can benefit from being flexible in their behaviour towards changing environments in order to survive and reproduce [1]. In cognitive ecology, behavioural flexibility is taken as an indicator of advanced cognition [1]. Social factors have often been used to explain more advanced cognition, while less focus has been placed on ecological factors [2]. To date, animal behavioural flexibility has been linked to complex social group structures [3,4] and to the diversity of food sources and habitats [5]. Another factor that has been argued to impact on behavioural flexibility is the level of domestication of a population (e.g. wolves (*Canis lupus*) are more flexible than domestic dogs (*Canis lupus familiaris*) [6]). Personality, i.e. stable inter-individual differences in behaviour, has also been shown to affect the degree of behavioural flexibility, although its direction is often confounded by many other factors [7]. However, the link between behavioural flexibility and ecological factors, such as foraging strategies [5], is not yet well established, especially in other taxa than primates. For instance, MacLean *et al*. [8] report that within primates, dietary breadth but not social group size is predictive of species differences in self-control. In the context of this study, we focussed on the two most commonly used operationalizations of behavioural flexibility in the context of animal cognition: the ability to control or inhibit certain behaviours [9], and reversal learning [10]. Inhibitory control, which is linked to self-control and motor self-regulation (e.g. [6]), captures an animal's ability to suppress a prepotent direct response in reaction to a stimulus in favour of a more effective yet more effortful behaviour.

Detour paradigms have often been used to study inhibitory control [11], in which a subject is confronted with a direct and obvious path to the goal that is blocked and must therefore be avoided. One often-used detour test is the cylinder reaching task (see [8]), in which a subject must suppress direct reaching towards a reward in a transparent tube in favour of approaching it through one of the more distant openings. Owing to the fact that species might not have an understanding of the solidity of transparent (artificial) materials [12], this set-up might not yield valid results on inhibitory control. Therefore, the results of studies using this particular set-up need to be interpreted with caution.

A spatial detour task, in which subjects have to navigate around an obstacle, and thus momentarily increase the relative distance between themselves and a reward, offers an alternative test of inhibitory control while maintaining ecological validity (as it can be encountered as part of everyday life), and without recourse to training. In addition to assessing spatial problem-solving and inhibitory control, this task can measure the individuals' capacity for reversal learning simply by changing the layout. In this arrangement, a perseveration error, also called the A-not-B error, occurs when a subject attempts to use a previously reinforced detour path despite a visible barrier shift that requires a new route [13]. This follows the same principle as a classic object permanence experiment where an object is repeatedly hidden in the A-location before visibly being hidden in the B-location. Here, perseveration errors in the search and reaching behaviour were observed in adult cats (*Felis catus*), adult dogs (*Canis lupus familiaris*) and grey parrots (*Psittacus erithacus*), but not in kittens, puppies or goats (*Capra aegagrus hircus*) (for a review see [14]). However, the lack of a unified testing procedure makes comparisons difficult.

The spatial A-not-B detour task approach of McKenzie & Bigelow [13], developed for infants, offers a set-up that is adaptable for all mobile species. Recent studies using this task found that dogs [15] and horses (*Equus caballus*) [16] persistently committed perseveration errors, whereas mules (*Equus asinus* x *Equus caballus*) and donkeys (*Equus asinus*) showed a higher level of behavioural flexibility [16]. The unexpected higher flexibility in donkeys and mules in comparison to horses in the spatial A-not-B set-up might be explained by their evolved responses to challenging situations: instead of flight they stand their ground, therefore inhibiting a flight response [17]. Dogs, as predators, probably possess different problem-solving abilities from those of prey animals (e.g. persistence in dogs [15,18]) yet their perseverance rates equal those of horses, a typical flight animal. These inter-species comparisons of cognitive capacities are useful to identify possible evolutionary reasons for differences, which can then be taken as inspiration for studies with additional species.

Goats and sheep are not only phylogenetic neighbours with a similar domestication history (they were domesticated about 10 500 years ago [19]), but also share similar social structures (complex fission–fusion societies [20,21]) and have been reported to have similar visual acuities (including a wide visual angle and a poor depth perception; for a review see [14]). However, they differ substantially in their foraging strategies [22]. Goats are browsers, foraging low-fibre vegetation from various heights (e.g. foliage, buds, flowers, and stems of shrubbery; i.e. feeding more patchily distributed food), while sheep are typical grazers, feeding on high-fibre herbaceous species (i.e. feeding more evenly distributed food [23–25]). Differences in foraging strategies are closely linked

to decision-making processes [2], which might therefore vary between goats (more flexible foraging strategy) and sheep (less flexible foraging strategy) (e.g. [26–28]). While there were no differences in some cognitive capacities between the species with regards to e.g. discrimination [29–31], learning and memory [32–34], and operant conditioning [35,36], Nawroth *et al.* [26] used differences in the feeding ecology of goats and sheep to explain differences in their exclusion performance. In their study, only goats used indirect information to infer the location of a hidden food reward (while both species were able to use direct information), thus demonstrating a more flexible search strategy. Both goats [37] and sheep [38] have been shown to self-regulate their behaviour in the above-mentioned cylinder detour task. Whether this self-regulation would be found in a spatial, and thus more ecologically valid, detour setting, needs to be explored. At the same time, we expected that the variation in feeding ecology would have an effect on the species' performance in a detour task.

As a measure of behavioural flexibility, we investigated here whether goats and sheep differ in their performance on a spatial A-not-B detour task. We expected for both species a peak in perseveration errors when the gap location was changed (first B trial), and that goats would overcome this perseveration faster than sheep owing to their more flexible foraging strategy. Additionally, and in accordance with findings in other species, the number of errors was expected to be related to the number of received A trials, i.e. the more experience subjects had with the A location, the higher and more persistent the perseveration rate would be.

# 2. Material and methods

## 2.1. Ethics statement, management conditions and subjects

Twenty-one goats (eight females and 13 males; seven pygmy, three Anglo Nubian, two British Saanen, one Boer, one Toggenburg and seven mixed breeds; ranging from 3 to 17 years) were used in the testing during summer 2016. The animals were part of a larger group that roamed free in a large grassy paddock during the day and were kept indoors overnight, either in small groups or individually. Hay, grass, and water were available ad libitum, as well as a commercial concentrate fed varying according to the goats' health condition and age. All the animals were used to being handled and many of them had participated in previous experiments (e.g. [40,41]).

Twenty-eight non-lactating and non-reproducing female sheep (25 Lacaune and three East Friesian sheep born between January and February 2018) were tested during the summer of 2018. They were housed from April 2018 in seven group pens (2.4 × 3.5 m each pen) in an open-front barn at the Agroscope Research Station in Tänikon, Switzerland. They had straw bedding available, hay was provided twice a day at a regular time and water was available ad libitum. All the animals were habituated to being handled by an experimenter (CMCR) and took part in previous experiments involving acoustic stimuli [42].

## 2.2. Experimental procedure

We followed the same methodology as Osthaus *et al.* [15,16]. In both locations, the testing arena—a large rectangular pen (5.3 × 7 m) with a straight barrier across the middle—was constructed from metal farmstock fencing elements (figure 1), either on grass (goats) or a concrete floor (sheep). Each animal was tested individually with only auditory contact with its herd members. Both groups were tested in a location familiar to them, by a familiar person, and received the reward from a previously used receptacle (bucket). During testing, the subject had to walk unaccompanied from a starting point at one end of the pen to a target (i.e. a familiar human with a small food reward) at the other end of the pen through a clearly visible gap in the barrier. The goats were reinforced with dried pasta while the sheep earned a small amount of a mixture of UFA 763 (ProRumin COMBI QM, Herzogenbuchsee, Switzerland). The starting location of the gap (left/right) was counterbalanced between subjects. After either one, two, three, or four A trial(s), the gap in the barrier was moved to the opposite side and each subject completed four B trials (i.e. B1, B2, B3 and B4). At the end of each trial, the subject was led back to the starting point around the outside of the pen, counterbalanced between left and right. Goats and sheep were pseudorandomly allocated to one of the four test conditions: one A trial only (i.e. A1; $n = 5$ goats and 7 sheep), two A trials (i.e. A1 and A2; $n = 5$ goats and 7 sheep), three A trials (i.e. A1, A2 and A3; $n = 5$ goats and 7 sheep) and four A trials (i.e. A1, A2, A3 and A4; $n = 6$

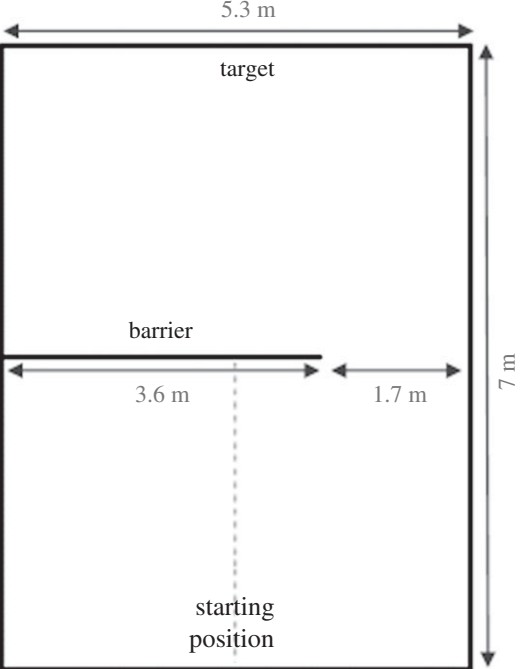

**Figure 1.** Layout and dimensions of the test arena (top view), with the gap on the right side of the barrier. The dashed vertical line represents a coding boundary that was used to define a correct or incorrect response.

goats and 7 sheep). Sex (for the goats), breed, and initial side of the gap were counterbalanced across conditions.

## 2.3. Measurements and data scoring

All trials were video-recorded (Sony HDR-CX190E for the goats and DCR-SX33E for the sheep; Camcorder, Sony Corporation, Tokyo, Japan). The performance was measured by accuracy (correct/incorrect) and latency. A trial was considered correct when the subject walked towards the gap without crossing the coding boundary in the middle of the pen (dashed verticle line in figure 1) and was incorrect when it moved into the area opposite the gap location. Latency was defined as the time taken from release at the starting point to crossing the barrier gap with the shoulders. If the subject had not passed the barrier within 2 min, the trial was stopped, and a latency of 120 s was recorded. For each trial, the performance indicators were recorded during testing with a stopwatch and a score sheet, and additionally two scorers measured the performance indicators over all trials based on video recordings. Data from the first scorer (CMCR) were used for the statistical analyses.

## 2.4. Statistical analysis

Statistical analyses were performed in R v. 3.6.3 [43]. The accuracy rates of goats and sheep for each A and B trial were compared to chance levels using two-sided binomial tests, and the differences between the species or conditions in accuracy rates were analysed with Fisher's exact tests. As latencies were not normally distributed (and could not be corrected), species latency differences were assessed using Mann–Whitney $U$-tests and the respective effect sizes were calculated. McNemar tests (calculating exact binomial probabilities) were used for each species to compare changes in accuracy rates from their last A trial to the first B trial. The number of subjects with positive or negative time differences between their last A trial to the first B trial across species were analysed with a Fisher's exact test. The effect of the number of A trial repetitions on the latencies in the B1 trial was determined using a Jonckheere–Terpstra test. The reliability between the two scorers (based on video recordings) over all trials was measured using the information-based measure of disagreement (IBMD = 0.10 [0.10; 0.11]) for the latency to pass the barrier through the gap, and using raw agreement indices (RAI = 0.97 [0.94; 0.98]) for the accuracy (package obs.agree [44]).

# 3. Results

## 3.1. Performance in A trials

Goats were more accurate than sheep in the first A trial (table 1). Except for sheep in A1, both species performed above-chance level in all A trials (table 1). In the fourth trial, both goats and sheep achieved 100% correct responses (table 1).

There were no latency differences between goats and sheep in A1 (Mann–Whitney $U$-test: $W = 288.5$, $p = 0.92$, $r = 0.02$; figure 2), while sheep were faster than goats to perform the detour in all other A trials (A2: $W = 237$, $p = 0.035$, $r = 0.35$; A3: $W = 152.5$, $p < 0.001$, $r = 0.83$; A4: $W = 42$, $p = 0.003$, $r = 0.84$).

## 3.2. Performance in B trials

After the gap location was moved to the opposite side of the barrier, the accuracy rates were reduced. In the first B trial, goats performed at chance level, while sheep performed below chance level, thus committing the perseveration error (table 1). No statistical difference in the species' performance was found in B1 (table 1). In B2 and B3, goats took the correct path above-chance level, while sheep performed at chance level: a statistical difference between species was observed for these trials (table 1; see also electronic supplementary material, video). In the last B trial, both goats and sheep performed above chance and did not differ in their performance (table 1).

The latencies in the first B trials did not statistically differ between both species (Mann–Whitney $U$-test: B1: $W = 224$, $p = 0.16$, $r = 0.2$), probably owing to the large variation in individual speeds (figure 2). Latencies between species also did not differ in the second B trial (B2: $W = 288$, $p = 0.91$, $r = 0.02$), but in the last two B trials, sheep were again faster than goats (B3: $W = 406.5$, $p = 0.03$, $r = 0.33$; B4: $W = 464.5$, $p < 0.001$, $r = 0.49$).

## 3.3. Effect of gap position change

Both species showed a reduction in their accuracy rates from a subject's last A trial to first B trial (McNemar's test: goats: $\chi_1^2 = 9$, $p = 0.003$; sheep: $\chi_1^2 = 16.67$, $p < 0.001$), and no difference between the two species was observed with regards to the number of subjects that improved or regressed (Fisher's exact test: $p > 0.99$). There was no difference in the effect of the change in gap location on latencies between species (Mann–Whitney $U$-test: $W = 224$, $p = 0.16$, $r = 0.2$), probably because of the large individual speed variations (from last A trial to first B trial in goats 5.60 [3.60; 9.50] versus sheep 10.15 [5.53; 20.60]).

## 3.4. Effect of the number of A trials on responses to the first B trial

Across both species, individuals that did only one A trial performed at chance level (McNemar's test: A1 $\chi_1^2 = 1.29$, $p = 0.26$), while individuals that did two, three or four A trials were more likely to commit the perseveration error (A2: $\chi_1^2 = 8$, $p = 0.005$; A3: $\chi_1^2 = 8$, $p = 0.005$; A4: $\chi_1^2 = 10$, $p = 0.002$). In both species, the latency to pass the barrier in B1 seemed to increase with the number of A trials (median [lower; upper quartile], one A trial: 5.25 [3.08; 8.08], two A trials: 7.55 [4.13; 18.70], three A trials: 9.05 [6.30; 20.30] and four A trials: 12.50 [5.70; 28.70]), though it was not supported statistically (Jonckheere–Terpstra's test: $k = 2$, $Z = 1.71$, $p = 0.087$).

# 4. Discussion

Goats and sheep are similar in a range of socio-ecological features but vary in their feeding ecology. It was expected that goats would be more flexible in their detour behaviour and reversal learning than sheep, in that goats were expected to recover faster from the perseveration error than sheep. Both species were able to complete a simple spatial detour task, although 40% of the sheep did not go straight to the visible gap in the first trial (all of the goats did so). After the change of gap location (i.e. first B trial), both goats and sheep showed a decrease in performance (i.e. a decrease in accuracy and an increase in latency times). The goats chose the correct side above chance level from their second B trial onwards, whereas the sheep achieved an above-chance performance only in B4. As hypothesised, goats overcame their perseverance faster than sheep and therefore showed a

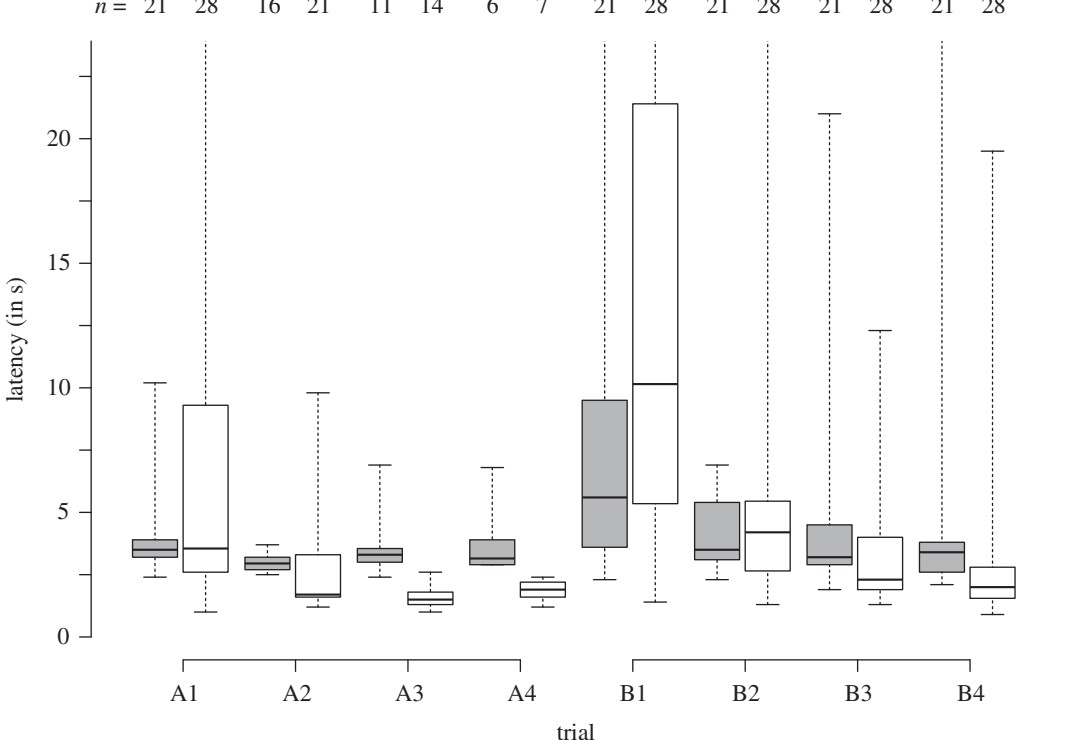

**Figure 2.** Latencies to perform the detour task in seconds, per species (goats in grey, sheep in white) and per trial. Boxplots indicate data range, median, as well as lower and upper quartiles; *n*: number of animals tested per trial and species.

**Table 1.** Detour task accuracy rates[a] and their *p*-values[b] per species and per trial.

| trial | goats | | sheep | | species comparison |
|---|---|---|---|---|---|
| | accuracy | *p*-value | accuracy | *p*-value | *p*-value |
| A1 | 21/21* | <0.0001 | 17/28 | 0.345 | 0.003 |
| A2 | 16/16* | <0.0001 | 19/21* | 0.0002 | 0.50 |
| A3 | 10/11* | 0.0117 | 14/14* | 0.0001 | 0.44 |
| A4 | 6/6* | 0.031 | 7/7* | 0.016 | — |
| B1 | 9/21 | 0.664 | 5/28[†] | 0.0009 | 0.11 |
| B2 | 16/21* | 0.0266 | 12/28 | 0.572 | 0.03 |
| B3 | 20/21* | <0.0001 | 19/28 | 0.087 | 0.03 |
| B4 | 21/21* | <0.0001 | 24/28* | 0.0002 | 0.13 |

[a]The number of subjects that moved towards the correct side of the barrier without deviation compared to the number of subjects tested per condition.
[b]As measured in each species by a two-sided binomial test with group performing at chance level (), above-chance level (*), or below chance level (†) and by Fisher's exact tests for the species comparisons.

higher level of flexibility. Our results further validate previous comparative studies that found goats to outperform sheep in cognitive tasks based on flexible decision-making (e.g. inferential reasoning [26], higher win-shift strategy [28]). Because both species share similar evolutionary and domestication histories [19], the ability of goats to adapt to a change in the spatial constellation of the task faster than sheep might be linked to the difference in foraging strategies between these two species. Behavioural flexibility is favoured in species exploiting diverse food sources or inhabiting environments with highly unpredictable resources [1]. Goats (browsers) rely on patchily distributed food sources that are less predictable than the more evenly distributed food sources that sheep

(grazers) rely on. It might be argued that the differences are caused by greater fearfulness in the sheep (faster speeds, less time to perceive and think). While there are differences in independence from conspecifics (for sheep it has been shown that they value proximity to conspecifics, so-called group bonds, higher than food quality (e.g. [45,46])), these differences are ultimately linked to feeding strategies: a more flexible browser needs to be comfortable at a distance from the group, unlike a grazer.

When comparing species response in an A-not-B detour task, it appears that the performances of the goats were similar to those of mules and donkeys [16], whereas sheep behaved in a manner more consistent with dogs [15] and horses [16], committing the perseveration error for longer. In sheep, dogs, horses and infants [13], habitual motor responses often overruled the sensory integration of the new context, whereas mules, donkeys and goats showed increased goal-directed information processing [47].

We observed that sheep generally walked faster than goats, except in the first A trial (see below) and the two first B trials (sheep latencies and performance rates were impaired by the change of gap location). As both species have approximately the same body size, this finding could highlight motivational issues in goats, as well as a higher motivation or heightened arousal in sheep. The type of reward, physiological state, and motivation have been found to affect subjects' detour response [48,49]. Here, the food rewards used for the two species differed (i.e. dried pasta for goats versus cereals mixture for sheep) but were each highly appealing, though animals were not food deprived for the experiment. Thus, a reasonable explanation would be that sheep were generally more motivated, either to retrieve the food faster (see also [38]) and/or to approach the familiar handler on the other side. In humans, it has also been shown that reward negatively affects inhibition [50] and therefore the difference in the attitude towards the reward might have affected the results. We also observed a large variation in our animals' individual speeds that could be linked to personality [45,46,51,52] or different breeds and sizes for the goats. Although previous studies in cylinder detour tasks or Y mazes reported that neither goats nor sheep displayed a lateral bias at the population level, they observed individual side biases (e.g. in goats, [37] and sheep, [53]), i.e. a consistent preference to detour to the left or right side, that might also have affected the accuracy results in the current study.

In our study, sheep might initially have shown some neophobia to the testing situation [48], or a higher level of explorative behaviour than goats, therefore impairing their latencies and accuracy rates in the first A trial compared to the following trials. The same effect of slow first trials was also observable in donkeys and horses [16], but not in dogs [15] or mules [16]. Other factors, such as age, sex, breed and personality (see also [11]), might have influenced the observed behaviours. While young subjects might fail because of motor control immaturity (e.g. infants under one year old [54]), response inhibition might decline with age (e.g. dogs from about eight years old [55]) and older animals might be slower to gather new information (see [56]). Although in our study, the sheep were younger (5–7 months old, i.e. close to sexual maturity) than the goats (6.5 years old in average), we do not assume that age might have altered the results. Hunter *et al*. [57] investigated the effects of age and experience on (reversal) learning and memory performances in a detour maze in sheep and observed no differences between 4- and 9-month-old naïve sheep. This indicates that the sheep used in our study had already reached cognitive maturity. Previous studies obtained conflicting results on how behavioural flexibility is related to sex (e.g. no sex effect in pheasants [12], but female bank voles are more flexible than males [56]), breed (e.g. in dogs [18], but see also [58]), personality (e.g. level of activity and fearfulness in zebra finches [59], proactive/reactive common waxbills [60]) or a combination (e.g. sex- and personality-dependent in great tit flexible learning abilities [61]). In the current study, we did not test for breed or personality effects, but we checked for potential sex differences in goats. Keeping in mind the sample size (i.e. low power statistical analyses), we found that male goats were faster than female goats in A1, A2 and A3 only (see the electronic supplementary material, R script), but no effect on accuracy rates in any of the A or B trials was detected. Thus, although sex did not seem to influence the accuracy rates in the current experiment, potential sex effect should be taken into account when designing future experiments (see [62]).

Moreover, we cannot rule out that site-specific idiosyncrasies (e.g. feeding and/or handling regimes) and general differences in the subjects' rearing histories might have had an impact on the general behaviour of the animals. However, logistics and resources did not enable us to test goats and sheep at the same site. Future research should take into account these confounding factors.

Across both species, repetitions of the A trial seemed to hinder the adoption of a novel correct path in subsequent B trials. The animals that received two, three or four A trials were more likely to commit the perseveration error. This conforms with previous observations in dogs [15] and equines [16].

# 5. Conclusion

We found that goats showed a faster recovery from the perseveration error compared to sheep (i.e. decreased perseveration errors) in a spatial detour task. These differences could be explained owing to the differences in foraging strategies between the two species [2]. Further research, e.g. by extending the range of species, will be necessary to better understand the impact of ecological factors, such as feeding strategies, on behavioural flexibility in animals, and to validate our findings using a variety of assays of behavioural flexibility. For domestic animals, these results could also help improve animal welfare guidelines by taking into account species-specific variabilities of inhibitory skills in husbandry practices, such as moving and handling. The design of future behavioural studies should also consider species-specific socio-ecological predisposition to allow for valid conclusions.

Ethics. Testing took place at two different locations: goats were tested at the Buttercups Sanctuary for Goats in Kent, UK (study approved by the Animal Welfare and Ethical Review Board Committee of Queen Mary University of London, ref. QMULAWERB072016), and sheep were tested at the Agroscope Research Station in Tänikon, Switzerland (research approved by the Research Commission of the Federal Food Safety and Veterinary Office, and necessary authorization to conduct animal experiments granted by the cantonal authorities, Canton of Thurgau permit no. 27508-TG01/16). Animal care and experimental procedures were in accordance with the ASAB/ABS Guidelines for Use of Animals in Research [39], and only positive reinforcement was used as a motivation during the experiments.

Data accessibility. The data and script underlying this study are available from the electronic supplementary material (video, raw data and R script).

Authors' contributions. C.M.C.R., B.O., A.G.M. and C.N. designed and conceived the study. C.M.C.R., B.O., A.C.G.H. and C.N. conducted the experiments. C.M.C.R. performed the statistical analyses. C.M.C.R., B.O. and C.N. interpreted the data and wrote the manuscript, and A.C.G.H. and A.G.M. made substantial contributions in revising the manuscript. All the authors gave their final approval for publication.

Competing interests. The authors have no competing interests.

Funding. This work was supported by grants from the Federal Food Safety and Veterinary Office (grant no. 2.16.05) to C.M.C.R., the Deutsche Forschungsgemeinschaft (NA 1233/1-1) to C.N., and Farm Sanctuary 'The Someone Project' to A.G.M. and C.N.

Acknowledgements. We would like to thank all staff and volunteers at Buttercups Sanctuary for Goats (www.buttercups.org.uk) for their help and free access to the goats, as well as Agroscope Tänikon for providing the sheep, Floriane Granger for helping with the testing of the sheep and the MLU Halle for the resources to develop this article.

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
