## [Peer Review File · Royal Society Open Science]

Review History

RSOS-201627.R0 (Original submission)

Review form: Reviewer 1

Is the manuscript scientifically sound in its present form?

No

Are the interpretations and conclusions justified by the results?

No

Is the language acceptable?

Yes

Do you have any ethical concerns with this paper?

No

Have you any concerns about statistical analyses in this paper?

No

Recommendation?

Reject

Comments to the Author(s)

Major issues:

The authors appear to pin-point "feeding ecology" as the most likely reason for their observed species differences in goat and sheep inhibition, but I do not agree. As the authors rightly discuss, detour tasks are not necessarily valid measures of inhibitory control and since the authors do not provide any evidence to show the underlying validity of their task, it's difficult to conclude that their performance data are indeed robust measures of that particular cognitive trait. Thus, while their data are potentially publishable, they must rework a lot of the paper to be more cautious about what exactly the authors have really measured in their animals (e.g. better discussion and consideration over possible confounds like species differences in vision, domestication, rearing history, food preferences, personality, risk-taking behaviour, etc.). I understand they've discussed this in their Discussion section, but they cannot reach their final conclusions due to those constraints. Firstly, there is no evidence in this paper that directly ties species variation in foraging to performance on this cognitive task. To do so would require further experimental manipulations that link, for example, individual variation in inhibition to differences in foraging strategies. Otherwise, it's impossible to tell whether any observed differences between species are due to something else unrelated to foraging (e.g. risk-taking travel behaviour in an unfamiliar enclosure). Secondly, the authors didn't actually find evidence for species differences that were consistent across conditions; in other words, the significant differences they report were only for some conditions, but the pattern did not necessarily make sense. Why, for example, would you jump to the conclusion that goats are "more flexible" than sheep in the B trials if they only differed in B2 and B3? What's special about B2 and B3, compared to B1 or B4 - why be more flexible sometimes but not on other occasions?

Minor Comments:

Line 34: It's better to say animals "can benefit" from being flexible since you don't always *need* to be flexible to survive and reproduce. For instance, if the environmental changes are an even better fit for the species' niche, there's no need to really change what you're already doing.

Line 42-43: How so? Explain now because it seems to be an important focal point later on.

Line 48-49: Who so? Explain. Unclear how it's "also a key factor".

Line 55-57: You just said on line 51 that detour tasks "demonstrate levels of inhibitory control" so you should reword either of these lines to be clearer in what you mean to say, which is that such tasks are "putative" measures of that trait.

Line 101: How is a barrier task *more* valid? If, for example, you're a sheep in an open field (with no barriers), then a barrier would be no more valid than a cylinder. You should clarify what you mean by providing a better rationale for why you think such a task is more valid for you animals specifically.

Lines 109-111: I agree with the prediction, but how is this relevant to your main question, which is about species differences?

Lines 114-123: Since your goats and sheep were tested in completely different settings, how might situational effects have impacted any apparent "species" differences? Also, without any food preference data, it's unclear whether species performed differently simply due to differences in food (and thus motivation to seek the reward).

Lines 155-158: The sample sizes do not reflect the ones you report in Table 1 and Figure 2, so please clarify exactly how many animals per species went into each condition. For instance, from my understanding of what you've written, there should have been 5 goats in A1, but Table 1 gives the impression that it was 21 goats in A1.

Figure 2: Why did you have variation in sample sizes across all your conditions, why not have equal numbers across? For instance, why have 21 goats in A1 but only 6 in A4? It otherwise gives the impression that there may be some underlying selection bias in these data, so that should be clarified if possible.

Line 265: Reference needed for evolutionary and domestication histories.

Lines 295-296: Explain what you mean here by "individual biases".

Lines 298-301: While I completely understand that studies are limited by the number of variables they can control for within a study, it is nevertheless very well known that neophobia and other situational/dispositional factors can impact latency to approach or perform a behaviour, so if, as the authors say, neophobia was a possible reason for species differences in the task, then why was this not taken into account in the study design (e.g. habituating subjects to the enclosure prior to testing)?

Line 308: If age is a concern, then could you not test for age effects with your data?

Line 320: Here you say sex did not influence the experiment, but on line 318 you say that it did in some respects. Please be clear what you mean.

Lines 325-326: Please explain the significance of this statement since, at present, it reads more like a random observation rather than something that's theoretically meaningful to your paper.

Line 330: Not true what you say -- goats were more flexible only for some (not all) conditions. Also, the conditions where goats were seemingly more flexible did not fall into any sort of pattern (e.g. increasing task complexity); instead, it appears to be random -- again, why for example, would they have been more flexible in B2 and B3 but not B1 and B4? Please discuss.

Lines 333-336: While true, this statement seems like a random afterthought and doesn't fit well with the rest of the conclusion section. Put that somewhere else in your discussion.

Review form: Reviewer 2

Is the manuscript scientifically sound in its present form?

Yes

Are the interpretations and conclusions justified by the results?

Yes

Is the language acceptable?

Yes

Do you have any ethical concerns with this paper?

No

Have you any concerns about statistical analyses in this paper?

No

Recommendation?

Accept as is

Comments to the Author(s)

This is a well designed study, with appropriate analysis and interpretation of the results. I find no fault with this study.

Decision letter (RSOS-201627.R0)

Dear Dr Nawroth

The Editors assigned to your paper RSOS-201627 "Goats show higher behavioural flexibility than sheep in a spatial detour task" have now received comments from reviewers and would like you to revise the paper in accordance with the reviewer comments and any comments from the Editors. Please note this decision does not guarantee eventual acceptance.

Please submit your revised manuscript and required files (see below) no later than 21 days from today's (ie 11-Dec-2020) date. Note: the ScholarOne system will 'lock' if submission of the revision is attempted 21 or more days after the deadline. If you do not think you will be able to meet this deadline please contact the editorial office immediately.

Best regards,
Lianne Parkhouse
Editorial Coordinator

on behalf of Dr Rosalind Arden (Associate Editor) and Kevin Padian (Subject Editor)
openscience@royalsociety.org

Subject Editor Comments to Author:

Thanks for your submission. As you see, there are two very different reviews. One reviewer approves of the paper but offers no substantial comment. The other feels that the inferences are unjustified given the number of possible variables that would need to be tested. Please address each of their comments in your response. We will try to send your revision to this reviewer for another look. Best wishes.

Associate Editor Comments to Author (Dr Rosalind Arden):

Thank you for submitting this work to RSOS. As often happens, two expert reviewers have different views on the work. We would very much like you to read the comments from Reviewer 1 and respond to them. A re-submission of your ms would be welcome if you would like to address the suggestions contained in that review. The comments made by Reviewer one seem reasonable, constructive and worth attending to. Reviewer 2's endorsement supports the contention that the work is very interesting.

Reviewer comments to Author:

Reviewer: 1
Comments to the Author(s)

Major issues:

The authors appear to pin-point "feeding ecology" as the most likely reason for their observed species differences in goat and sheep inhibition, but I do not agree. As the authors rightly discuss, detour tasks are not necessarily valid measures of inhibitory control and since the authors do not provide any evidence to show the underlying validity of their task, it's difficult to conclude that their performance data are indeed robust measures of that particular cognitive trait. Thus, while their data are potentially publishable, they must rework a lot of the paper to be more cautious about what exactly the authors have really measured in their animals (e.g. better discussion and consideration over possible confounds like species differences in vision, domestication, rearing history, food preferences, personality, risk-taking behaviour, etc.). I understand they've discussed this in their Discussion section, but they cannot reach their final conclusions due to those constraints. Firstly, there is no evidence in this paper that directly ties species variation in foraging to performance on this cognitive task. To do so would require further experimental manipulations that link, for example, individual variation in inhibition to differences in foraging strategies. Otherwise, it's impossible to tell whether any observed differences between species are due to something else unrelated to foraging (e.g. risk-taking travel behaviour in an unfamiliar enclosure). Secondly, the authors didn't actually find evidence for species differences that were consistent across conditions; in other words, the significant differences they report were only for some conditions, but the pattern did not necessarily make sense. Why, for example, would you jump to the conclusion that goats are "more flexible" than sheep in the B trials if they only differed in B2 and B3? What's special about B2 and B3, compared to B1 or B4 - why be more flexible sometimes but not on other occasions?

Minor Comments:

Line 34: It's better to say animals "can benefit" from being flexible since you don't always *need* to be flexible to survive and reproduce. For instance, if the environmental changes are an even better fit for the species' niche, there's no need to really change what you're already doing.

Line 42-43: How so? Explain now because it seems to be an important focal point later on.

Line 48-49: Who so? Explain. Unclear how it's "also a key factor".

Line 55-57: You just said on line 51 that detour tasks "demonstrate levels of inhibitory control" so you should reword either of these lines to be clearer in what you mean to say, which is that such tasks are "putative" measures of that trait.

Line 101: How is a barrier task *more* valid? If, for example, you're a sheep in an open field (with no barriers), then a barrier would be no more valid than a cylinder. You should clarify what you mean by providing a better rationale for why you think such a task is more valid for you animals specifically.

Lines 109-111: I agree with the prediction, but how is this relevant to your main question, which is about species differences?

Lines 114-123: Since your goats and sheep were tested in completely different settings, how might situational effects have impacted any apparent "species" differences? Also, without any food preference data, it's unclear whether species performed differently simply due to differences in food (and thus motivation to seek the reward).

Lines 155-158: The sample sizes do not reflect the ones you report in Table 1 and Figure 2, so please clarify exactly how many animals per species went into each condition. For instance, from my understanding of what you've written, there should have been 5 goats in A1, but Table 1 gives the impression that it was 21 goats in A1.

Figure 2: Why did you have variation in sample sizes across all your conditions, why not have equal numbers across? For instance, why have 21 goats in A1 but only 6 in A4? It otherwise gives the impression that there may be some underlying selection bias in these data, so that should be clarified if possible.

Line 265: Reference needed for evolutionary and domestication histories.

Lines 295-296: Explain what you mean here by "individual biases".

Lines 298-301: While I completely understand that studies are limited by the number of variables they can control for within a study, it is nevertheless very well known that neophobia and other situational/dispositional factors can impact latency to approach or perform a behaviour, so if, as the authors say, neophobia was a possible reason for species differences in the task, then why was this not taken into account in the study design (e.g. habituating subjects to the enclosure prior to testing)?

Line 308: If age is a concern, then could you not test for age effects with your data?

Line 320: Here you say sex did not influence the experiment, but on line 318 you say that it did in some respects. Please be clear what you mean.

Lines 325-326: Please explain the significance of this statement since, at present, it reads more like a random observation rather than something that's theoretically meaningful to your paper.

Line 330: Not true what you say -- goats were more flexible only for some (not all) conditions. Also, the conditions where goats were seemingly more flexible did not fall into any sort of pattern (e.g. increasing task complexity); instead, it appears to be random -- again, why for example, would they have been more flexible in B2 and B3 but not B1 and B4? Please discuss.

Lines 333-336: While true, this statement seems like a random afterthought and doesn't fit well with the rest of the conclusion section. Put that somewhere else in your discussion.

Reviewer: 2

Comments to the Author(s)

This is a well designed study, with appropriate analysis and interpretation of the results. I find no fault with this study.

===PREPARING YOUR MANUSCRIPT===

===PREPARING YOUR REVISION IN SCHOLARONE===

Author's Response to Decision Letter for (RSOS-201627.R0)

See Appendix A.

Decision letter (RSOS-201627.R1)

Dear Dr Nawroth,

It is a pleasure to accept your manuscript entitled "Goats show higher behavioural flexibility than sheep in a spatial detour task" in its current form for publication in Royal Society Open Science.

on behalf of Dr Rosalind Arden (Associate Editor) and Kevin Padian (Subject Editor)
openscience@royalsociety.org

Associate Editor Comments to Author (Dr Rosalind Arden):

Thank you for the resubmission. Reviewer One had provided a wealth of comments (and Reviewer Two endorsed the work). Your responses to those helpful comments have strengthened the ms.

It is blindingly obvious that species differ in many ways including in cognition and flexibility, yet is also obvious that each species solves it's own particular relevant set of problems. Your work

here takes a well thought out first step in attempting to assess behavioural flexibility - and this work is theoretically coherent - foraging strategies could contribute to differences among species that share many ecological similarities.

I suggest that you include the Ns in Table 1 (for those readers who skim the paper and skipped over the abstract), but otherwise this makes a very welcome contribution to how we could think about between-species differences. Thank you for submitting to Royal Society Open Science.

Appendix A

Dear Dr. Padian, Dear Dr. Arden,

Thank you very much for the thoughtful and constructive comments by you and the two reviewers on our manuscript “Goats show higher behavioural flexibility than sheep in a spatial detour task” [RSOS-201627], and for giving us the chance of a resubmission. We have responded to all comments and highlighted the modifications in the manuscript in yellow.

As suggested by Reviewer 1, we added further relevant literature to strengthen our argument (i.e., species differences in behavioural flexibility can arise from different foraging strategies, L35-38, L42-46 and L273-276) and provided further support (in the point-by-point answers and the manuscript). We now also discuss possible confounding factors (such as rearing history and research site L334-338). The implication of the feedback has now significantly improved our manuscript and we hope that you will reconsider this manuscript for publication in Royal Society Open Science.

The material in this manuscript has not been published elsewhere and is not submitted for publication elsewhere. All authors have seen the final manuscript and we all take responsibility for its contents.

Yours sincerely,

C.M.C Raoult, B. Osthaus, A.C.G. Hildebrand, A.G. McElligott and C. Nawroth

Response to editors and reviewers

Subject Editor (Dr Kevin Padian) Comments to Author:

Thanks for your submission. As you see, there are two very different reviews. One reviewer approves of the paper but offers no substantial comment. The other feels that the inferences are unjustified given the number of possible variables that would need to be tested. Please address each of their comments in your response. We will try to send your revision to this reviewer for another look. Best wishes.

AU: Thank you for the positive feedback!

Associate Editor (Dr Rosalind Arden) Comments to Author:

Thank you for submitting this work to RSOS. As often happens, two expert reviewers have different views on the work. We would very much like you to read the comments from Reviewer 1 and respond to them. A re-submission of your ms would be welcome if you would like to address the suggestions contained in that review. The comments made by Reviewer one seem reasonable, constructive and worth attending to. Reviewer 2's endorsement supports the contention that the work is very interesting.

AU: Thank you for the positive feedback!

Response to Reviewer 1

Major issues:

The authors appear to pin-point "feeding ecology" as the most likely reason for their observed species differences in goat and sheep inhibition, but I do not agree. As the authors rightly discuss, detour tasks are not necessarily valid measures of inhibitory control and since the authors do not provide any evidence to show the underlying validity of their task, it's difficult to conclude that their performance data are indeed robust measures of that particular cognitive trait.

Thus, while their data are potentially publishable, they must rework a lot of the paper to be more cautious about what exactly the authors have really measured in their animals (e.g. better discussion and consideration over possible confounds like species differences in vision, domestication, rearing history, food preferences, personality, risk-taking behaviour, etc.). I understand they've discussed this in their Discussion section, but they cannot reach their final conclusions due to those constraints.

Firstly, there is no evidence in this paper that directly ties species variation in foraging to performance on this cognitive task. To do so would require further experimental manipulations that link, for example, individual variation in inhibition to differences in foraging strategies. Otherwise, it's impossible to tell whether any observed differences between species are due to something else unrelated to foraging (e.g. risk-taking travel behaviour in an unfamiliar enclosure).

AU: We would like to thank you for your detailed feedback. In the revised version of the manuscript, we addressed your comments and suggestions. Please see our replies to your items point-by-point below.

We think that the specific spatial detour task we used in this study is a valid measure to test behavioural flexibility. To clarify the matter, we have modified the paragraph in which we explain that certain detour paradigms (and the cylinder task in particular) cannot be used to study inhibitory control in animals (L53-60): "Detour paradigms have often been used to study inhibitory control [12], in which a subject is confronted with a direct and obvious path to the goal that is blocked and must therefore be avoided. One often used detour test is the cylinder reaching task [see 8], in which a subject must suppress a direct reaching towards a reward in a transparent tube in favour of approaching it through one of the more distant openings. Due to the fact that species might not have an understanding of the solidity of transparent (artificial) materials [13], this set-up might not yield valid results on inhibitory control. Therefore, the results of studies using this particular set-up need to be interpreted with caution". We also added a sentence in the Discussion section (L273-276): "Behavioural flexibility is favoured in species exploiting diverse food sources or inhabiting environments with highly unpredictable resources [1]. Goats (browsers) rely on patchily distributed food sources that are less predictable than the more evenly distributed food sources that sheep (grazers) rely on" and reworded a sentence in the abstract (L19-20): "We investigated whether goats' reliance on more patchily distributed food sources predicts an increased behavioural flexibility compared to sheep".

As suggested, we adjusted our conclusions (see L25-28, L261 and L346-351), in line with the potentially confounding factors, and provide further support/references where needed.

We specify (L90-93) that goats and sheep were domesticated about 10,500 years ago, are phylogenetic neighbours and have similar visual acuities: “Goats and sheep are not only phylogenetic neighbours with a similar domestication history [they were domesticated about 10,500 years ago, 19], but also share similar social structures [complex fission-fusion societies, 20,21] and have been reported to have similar visual acuities (including a wide visual angle and a poor depth perception; for a review see [22])”. We also acknowledge in the discussion section that goats and sheep were from – and tested on – two different sites and had different rearing history that potentially could have influenced the general response of the animals and should therefore be taken into account in future studies (L334-338): “Moreover, we cannot rule out that site-specific idiosyncrasies (e.g., feeding and/or handling regimes) and general differences in the subjects’ rearing histories might have had an impact on the general behaviour of the animals. However, logistics and resources did not enable us to test goats and sheep at the same site. Future research should take into account these confounding factors”. As regards to food preferences, there is no reason to assume that a lack of motivation would lead the animals to perform in the way we observed: both goats and sheep were able to complete a simple spatial detour task (A trials) and showed a clear drop in performance between A and B trials. Though we did not perform food preference tests, goats and sheep were provided with their putative favourite food each. Besides, both goats and sheep were motivated enough to detour the barrier in each single trial. Concerning the subjects’ personality, as all animals included were not pre-sectioned nor tested for specific personality traits, it is true that we did not assess, in the present study, the influence of personality on the spatial detour task performances (acknowledged L326-328). This could be included in a future study, though previous studies in other species reported conflicting results (see L322-326). Regarding risk-taking travel behaviour as an alternative explanation, there was, in our view, no (perceived) risk involved in our spatial detour task – subjects were always (food-)rewarded and went straight to the container with the rewards after passing the gap – so we cannot make any judgment on how goats and sheep would have perceived this environment as more or less risky.

Secondly, the authors didn't actually find evidence for species differences that were consistent across conditions; in other words, the significant differences they report were only for some conditions, but the pattern did not necessarily make sense. Why, for example, would you jump to the conclusion that goats are "more flexible" than sheep in the B trials if they only differed in B2 and B3? What's special about B2 and B3, compared to B1 or B4 - why be more flexible sometimes but not on other occasions?

AU: We have now provided a more detailed interpretation that better mirrors our data and the observed effects. Overall, we found that both species were able to complete a simple spatial detour task. However, when the gap location was changed, goats outperformed sheep in their accuracy rates as they overcome their perseveration faster (in fewer B trials) than sheep. After the change of the gap location (B trials), goats detoured the barrier accurately at chance level in the first B trial and above chance level from B2 onwards, whereas sheep performed below chance level in B1, at chance level in B2 and B3 and finally above chance level in B4. In short, we observed that goats recovered faster from the perseveration error than sheep. In B1, both goats and sheep made errors (no statistical difference). In B2 and B3, the goats’ performance levels recovered, but the sheep’s did not (statistical differences observed). In B4, both goats and sheep recovered (no statistical difference, a ceiling effect was likely reached for both goats and sheep). Differences are therefore not random but signal recovery from perseveration error. We rephrased our conclusion in the abstract (L25-28): “While both species showed an

accuracy reduction following the change of gap position, goats recovered from this perseveration error from the second B trial onwards, whereas sheep did so only in the fourth B trial, indicating differences in behavioural flexibility between the species”.

Minor Comments:

Line 34: It's better to say animals "can benefit" from being flexible since you don't always *need* to be flexible to survive and reproduce. For instance, if the environmental changes are an even better fit for the species' niche, there's no need to really change what you're already doing.

AU: As suggested, we revised the sentence (L33-34): “Animals can benefit from being flexible in their behaviour towards changing environments in order to survive and reproduce [1]”.

Line 42-43: How so? Explain now because it seems to be an important focal point later on.

AU: To date, most research on factors that might have shaped cognitive traits have focused on social factors, while ecological ones are equally important. We now further explain why we think the link between behavioural flexibility and foraging strategies needs, in our view, a deeper understanding L35-38: “Social factors have often been used to explain more advanced cognition, while less focus has been placed on ecological factors [2]). To date, animal behavioural flexibility has been linked to complex social group structures [3,4] and to the diversity of food sources and habitats [5]” and L42-46: “However, the link between behavioural flexibility and ecological factors, such as foraging strategies [5], is not yet well established, especially in other taxa than primates. For instance, MacLean et al. [8] report that within primates, dietary breadth but not social group size is predictive of species differences in self-control”.

Line 48-49: Who so? Explain. Unclear how it's "also a key factor".

AU: Because this sentence was redundant, we deleted it.

Line 55-57: You just said on line 51 that detour tasks "demonstrate levels of inhibitory control" so you should reword either of these lines to be clearer in what you mean to say, which is that such tasks are "putative" measures of that trait.

AU: We have now rephrased this section to clarify that the cylinder reaching task might not be an adequate measure for inhibitory control in animals (L53-60): “Detour paradigms have often been used to study inhibitory control [12], in which a subject is confronted with a direct and obvious path to the goal that is blocked and must therefore be avoided. One often used detour test is the cylinder reaching task [see 8], in which a subject must suppress direct reaching towards a reward in a transparent tube in favour of approaching it through one of the more distant openings. Due to the fact that species might not have an understanding of the solidity of transparent (artificial) materials [13], this setup might not yield valid results on inhibitory control. Therefore, the results of studies using this particular setup need to be interpreted with caution”.

Line 101: How is a barrier task *more* valid? If, for example, you're a sheep in an open field (with no barriers), then a barrier would be no more valid than a cylinder. You should clarify what you mean by providing a better rationale for why you think such a task is more valid for you animals specifically.

AU: A non-artificial barrier task is likely to be more ecologically valid than a transparent cylinder task because every mobile animal species encounters barriers in their natural world (barrier as in visible blockage of intended path). We clarify this in L53-60 (see response to the above question); L62-65: “A spatial detour task, in which subjects have to navigate around an obstacle, and thus momentarily increase the relative distance between themselves and a reward, offers an alternative test of inhibitory control while maintaining ecological validity (as it can be encountered as part of everyday life), and without recourse to training”; and L106-109: “Whether this self-regulation [observed in the cylinder task] would be found in a spatial, and thus more ecologically valid, detour setting, needs to be explored”.

Lines 109-111: I agree with the prediction, but how is this relevant to your main question, which is about species differences?

AU: We have now reworded this to show that this is a confirmatory hypothesis, in relation to other domesticated species that have already been tested with this setup (L115-116): “Additionally, and in accordance with findings in other species, the number of errors was expected to be related to the number of A trials, i.e. the more experience subjects had with the A location, the higher and more persistent the perseveration rate would be”.

Lines 114-123: Since your goats and sheep were tested in completely different settings, how might situational effects have impacted any apparent "species" differences? Also, without any food preference data, it's unclear whether species performed differently simply due to differences in food (and thus motivation to seek the reward).

AU: Although the experiments took place in two locations, the test design and arena size were the same for both species and all the animals had previous experiences with being handled and rewarded with either dry pasta (goats) or cereals-mix (sheep). The comparisons were made on the accuracy and latency, just as it was done in previous interspecies trials (donkeys, horses, mules; Osthaus et al., 2013). However, we cannot rule out that site-specific idiosyncrasies might have had an impact on the general behaviour of the animals. Unfortunately, in the current study, we were unable to test goats and sheep at the same site: it is extremely rare to have very similar groups of related species, accounting for habituation, age and the many other potential factors. Future research should be designed to overcome this limitation and thus further validate our findings. We now added a statement to our discussion to acknowledge this (L334-338): “Moreover, we cannot rule out that site-specific idiosyncrasies (e.g., feeding and/or handling regimes) and general differences in the subjects’ rearing histories might have had an impact on the general behaviour of the animals. However, logistics and resources did not enable us to test goats and sheep at the same site. Future research should take into account these confounding factors”.

In terms of food preference, there is no reason to assume that a lack of motivation would lead the animals to perform in the way we observed: both goats and sheep are highly motivated by their preferred food, and were able to complete a simple spatial detour task (A trials) and showed a clear drop in performance between A and B trials, despite the fact that they received the same food reward. It is clear from the results that both goats and sheep were motivated enough to detour the barrier. Please see also our argumentation related to motivation in the Discussion (L290-301): “We observed that sheep generally walked faster than goats, except in the first A trial (see below) and the two first B trials (sheep latencies and performance rates were impaired by the change of gap location). As both species have approximately the same body size, this finding could highlight motivational issues in goats, in addition to better

motivation or heightened arousal in sheep. The type of reward, physiological state, and motivation have been found to affect subjects' detour response [49,50]. Here, the food rewards used for the two species differed (i.e. dried pasta for goats vs. cereals mixture for sheep) but were each highly appealing, though animals were not food deprived for the experiment. Thus, a reasonable explanation would be that sheep were generally more motivated, either to retrieve the food faster [see also 39] and/or to approach the familiar handler on the other side. In humans, it has also been shown that reward negatively affects inhibition [51] and therefore the difference in the attitude towards the reward might have affected the results."

Lines 155-158: The sample sizes do not reflect the ones you report in Table 1 and Figure 2, so please clarify exactly how many animals per species went into each condition. For instance, from my understanding of what you've written, there should have been 5 goats in A1, but Table 1 gives the impression that it was 21 goats in A1.

AU: We have now clarified this: (L160) "each subject completed four B trials (i.e., B1, B2, B3 and B4)" and (L162-166) "Goat and sheep were pseudorandomly allocated to one of the four test conditions: one A trial (i.e., A1 only; n = 5 goats and 7 sheep), two A trials (i.e., A1 and A2; n = 5 goats and 7 sheep), three A trials (i.e., A1, A2 and A3; n = 5 goats and 7 sheep), and four A trials (i.e., A1, A2, A3 and A4; n = 6 goats and 7 sheep)".

In total, 21 goats and 28 sheep were tested. From these animals, approximately one quarter (i.e., 5 goats and 7 sheep) received only one A trial (A1) and 4 B trials (B1-B2-B3-B4), one quarter received only two A trials (A1 and A2) and 4 B trials, one quarter received only three A trials (A1-A2-A3) and 4 B trials, and the rest of the animals had four A trials (A1-A2-A3-A4) and 4 B trials. Therefore, all the animals involved performed at least A1, i.e., 21 goats and 28 sheep.

Figure 2: Why did you have variation in sample sizes across all your conditions, why not have equal numbers across? For instance, why have 21 goats in A1 but only 6 in A4? It otherwise gives the impression that there may be some underlying selection bias in these data, so that should be clarified if possible.

AU: The number of animals in the groups receiving 1, 2, 3 or 4 A trials in total were (almost) equal across one species. In Figure 2, we refer to the total number of animals receiving each trial, i.e., it also sums up subjects from the A2-A4 groups for A1). As all animals with 4 A trials will have done A1, A2 and A3 their data for those trials were include in the analysis. As these are consecutive trials, the later trials did not affect the performance on the earlier trials, i.e. an A4-condition animal would have the same conditions in the first trial as an A1-trial animal. No underlying selection of the animals was performed as they were "pseudo-randomly allocated to one of the four test conditions" (L162).

Line 265: Reference needed for evolutionary and domestication histories.

AU: We now refer to Alberto et al. (2018), (L271): "[19]"

Alberto FJ et al. 2018. Convergent genomic signatures of domestication in sheep and goats. Nature Communications 9, 813. (<http://dx.doi.org/10.1038/s41467-018-03206-y>).

Lines 295-296: Explain what you mean here by "individual biases".

AU: Animals have been observed to have lateralisation biases, i.e., a preference to perform behaviours by one side of the body or to process information differentially in the brain. In our study context, subjects could have exhibited a consistent preference to detour to the left or right

side. We rephrased (L303-306): “And although previous studies in cylinder detour tasks or Y mazes reported that neither goats nor sheep displayed a lateral bias at the population level, they observed individual side biases [e.g. in goats, 38, and sheep, 54], i.e., a consistent preference to detour to the left or right side, that might also have affected the accuracy results in the current study”.

Lines 298-301: While I completely understand that studies are limited by the number of variables they can control for within a study, it is nevertheless very well known that neophobia and other situational/dispositional factors can impact latency to approach or perform a behaviour, so if, as the authors say, neophobia was a possible reason for species differences in the task, then why was this not taken into account in the study design (e.g. habituating subjects to the enclosure prior to testing)?

AU: As we explained L152-153: “Both groups were tested in a location familiar to them, by a familiar person, and received the [known food] reward from a previously used receptacle (bucket)”. Besides, both the sheep and the goats were used to be handled in various settings. Moreover, to be able to compare our results to studies involving other animal species (e.g. dogs, equines), we had to use the same study design, i.e., not provide extensive enclosure/setup habituation. We made small modifications to the sentence (L308-311): “In our study, sheep might initially have shown some neophobia to the testing situation [49], or a higher level of explorative behaviour than goats, therefore impairing their latencies and accuracy rates in the first A trial compared to the following trials. The same effect of slow first trials was also observable in donkeys and horses [16], but not in ~~goats~~, dogs [15] or mules [16]”.

Line 308: If age is a concern, then could you not test for age effects with your data?

AU: With the current data and animals tested, we could not test for age effects. Though there was an age difference with sheep being on average younger than goats, at least all animals had reached cognitive maturity by the time of testing (L317-319): “Although in our study, the sheep were younger (5-7 months old, i.e. close to sexual maturity) than the goats (6.5 years old in average), we do not assume that age might have altered the results”. Ideally, we agree, testing for age effects would have alleviate any doubt. Again, it is extremely rare to have very similar groups of related species, accounting for habituation, age and the many other potential factors.

Line 320: Here you say sex did not influence the experiment, but on line 318 you say that it did in some respects. Please be clear what you mean.

AU: We made some modifications to the text to clarify our findings (L328-332): “Though the sample size was small (i.e., low power statistical analyses), male goats were found to be faster than female goats in A1, A2, and A3 only (see SEM), but no effect on accuracy rates in any of the A or B trials was detected. Thus, although sex did not seem to influence the accuracy rates in the current experiment, potential sex effect should be taken into account when designing future experiments [see 63]”.

Lines 325-326: Please explain the significance of this statement since, at present, it reads more like a random observation rather than something that's theoretically meaningful to your paper.

AU: This has now been reworded (L346-351): “We found that goats showed an improved recovery from the perseveration error compared to sheep (i.e., decreased perseveration errors) in a spatial detour task. These differences could be explained due to the differences in foraging strategies between the two species [2]. Further research, e.g. by extending the range of species,

will, however, be necessary to better understand the impact of ecological factors, such as feeding strategies, on behavioural flexibility in animal species, and to validate our findings using a variety of assays of behavioural flexibility” and its relevance for the rest of the paper has been enhanced.

Line 330: Not true what you say -- goats were more flexible only for some (not all) conditions. Also, the conditions where goats were seemingly more flexible did not fall into any sort of pattern (e.g. increasing task complexity); instead, it appears to be random -- again, why for example, would they have been more flexible in B2 and B3 but not B1 and B4? Please discuss.

AU: We hope that the interpretation of our results is now clearer due to the changes we added in the method section (L158-165): “After either one, two, three, or four A trial(s), the gap in the barrier was moved to the opposite side and each subject completed four B trials (i.e., B1, B2, B3 and B4). At the end of each trial, the subject was led back to the starting point by around the outside of the pen, counterbalanced between left and right. Goat and sheep were pseudo-randomly allocated to one of the four test conditions: one A trial only (i.e., A1; n = 5 goats and 7 sheep), two A trials (i.e., A1 and A2; n = 5 goats and 7 sheep), three A trials (i.e., A1, A2 and A3; n = 5 goats and 7 sheep), and four A trials (i.e., A1, A2, A3 and A4; n = 6 goats and 7 sheep)”. All B trials need to be seen in context as they are in a strict temporal order and indicate learning behaviour. The change over the four B trials indicates the flexibility: the later a species reaches the above-chance level, the less flexible it is. Goats reach their pre-change performance levels significantly faster than sheep. In other words, we observed that goats recovered faster from error than sheep. In B1, both goats and sheep made errors (no statistical difference). In B2 and B3, goats recovered, sheep did not (statistical differences observed). In B4, both goats and sheep recovered (no statistical difference, a ceiling effect was likely reached for both goats and sheep). Differences are therefore not random but signal recovery from perseveration error.

Lines 333-336: While true, this statement seems like a random afterthought and doesn't fit well with the rest of the conclusion section. Put that somewhere else in your discussion.

AU: We revised this sentence and made it fit better with the rest of the conclusion (L351-355): “For domestic animals, these results could also help improve animal welfare guidelines by taking into account species-specific variabilities of inhibitory skills in husbandry practices, such as moving and handling. The design of future behavioural studies should also consider species-specific socio-ecological predisposition to allow for valid conclusions”. Animal welfare was not the focus of our study but, as our findings are potentially relevant to it, we judged the conclusion to be the best location for this statement.

Response to Reviewer 2

This is a well designed study, with appropriate analysis and interpretation of the results. I find no fault with this study.

AU: Thank you very much for your endorsement.